# Current Status on Canine Foetal Fluid and Adnexa Derived Mesenchymal Stem Cells

**DOI:** 10.3390/ani11082254

**Published:** 2021-07-30

**Authors:** Eleonora Iacono, Romina Marcoccia, Barbara Merlo

**Affiliations:** Department of Veterinary Medical Sciences, University of Bologna, 40126 Bologna, Italy; romina.marcoccia2@unibo.it (R.M.); barbara.merlo@unibo.it (B.M.)

**Keywords:** dog, foetal adnexa, mesenchymal stem cells, therapy

## Abstract

**Simple Summary:**

In the last few years, dog owners have required sophisticated new treatments such as the use of MSCs for tissue engineering and regenerative medicine applications. On these topics, *Canis familiaris*, which develop many diseases with etiologies and pathogenesis similar to those that develop in humans, can be considered as a realistic preclinical model to evaluate the therapeutic potential of MSCs. The aim of the present review is to offer an update on the state of the art on canine MSCs derived from foetal adnexa and fluid, focusing on the findings in their clinical setting.

**Abstract:**

Effective standards of care treatment guidelines have been developed for many canine diseases. However, a subpopulation of patients is partially or completely refractory to these protocols, so their owners seek novel therapies such as treatments with MSCs. Although in dogs, as with human medicine, the most studied MSCs sources have been bone marrow and adipose tissue, in recent years, many researchers have drawn attention towards alternative sources, such as foetal adnexa and fluid, since they possess many advantages over bone marrow and adipose tissue. Foetal adnexa and fluid could be considered as discarded material; therefore, sampling is non-invasive, inexpensive and free from ethical considerations. Furthermore, MSCs derived from foetal adnexa and fluid preserve some of the characteristics of the primitive embryonic layers from which they originate and seem to present immune-modulatory properties that make them a good candidate for allo- and xenotransplantation. The aim of the present review is to offer an update on the state of the art on canine MSCs derived from foetal adnexa and fluid focusing on the findings in their clinical setting.

## 1. Introduction

Today, pet owners require their companion animals to be cured in veterinary hospitals, constantly monitored, using high diagnostic and medical devices. Furthermore, an increasing number of pet owners require their animals to be treated with sophisticated and new treatments. In this context, research on canine MSCs is becoming increasingly important.

On the other hand, because of the need for clinical trials in animal models for new treatments for human regenerative medicine and because humans and dogs share environmental life patterns and similar pathologies, *Canis familiaris* could be considered as a suitable model of spontaneous diseases [1,2,3,4,5,6].

For example, the most frequent and important non ischaemic cardiomyopathies, both in human and dog, showing pathological and clinical similarities, are dilated cardiomyopathy and arrhythmogenic ventricular cardiomyopathy [5]. DCM is the second most common cardiac disease, affecting a wide range of breeds, particularly Doberman Pinscher, with a prevalence of around 44% [6], while AVC has been described frequently in Boxer [7]. However, in both species, although progress in the management of symptoms has been made, the actual disease processes remain a challenge to treat, and study of the use of regenerative therapies using MSCs is very interesting.

Although the standing hindlimb angle, an aspect to consider for comparing biomechanical data, is much larger in canine than in human, dog could be an important model for osteoarthritis [8]. Indeed, compared with horse, ruminants or swine, using dog as a clinical model leads to easier post-surgical management and follow up [8]. In this context, dog can be considered the best model for SCI, cranial cruciate and meniscal injury, osteoarthritis and other skeletal muscle pathologies, both for studying mechanisms of degeneration and testing new treatments [6].

In addition, for gastro-intestinal pathologies, dog can be considered as an important clinical model, with the potential to overcome some of the major obstacles of laboratory animal modeling. Indeed, canine spontaneous lymphocytic-plasmocytic colitis, the most common form of enteropathy, included in inflammatory bowel disease syndrome, has several histopathologic and cellular-molecular features similar to human IBD [9]. This is characterized by multifactorial pathogenesis which is less compelling in rodent models [10], so dog can be essential for understanding MSCs immune modulation mechanisms, determining dose equivalence as well as biological effects of MSCs transplantation in patients affected by IBD refractory and traditional therapy [10].

Despite the importance of using the dog as model for human medicine, this raises more ethical debates than livestock animals. For this reasons, in the last decade, most studies in these animals have involved clinical cases of spontaneous pathologies observed in veterinary hospitals and clinics, highlighting their importance for clinical research [6].

To date, canine adult tissue, BM and AT have represented the most important sources of MSCs in the field of cell-based therapy [11,12,13] although cell harvesting is invasive, and increasing donor site morbidity and cell amount and characteristics are closely related to donor age [14,15,16,17,18,19]. Foetal fluids (amniotic fluid, umbilical cord blood), and foetal adnexa (Wharton’s jelly, amniotic membrane) have been identified as ideal alternative sources of MSCs in different animal species, such as horse [20,21,22], bovine [23,24], goat [25,26], and others. The benefits of these cells compared to adults MSCs and embryonic SCs are determined by their origin from extraembryonic tissues, usually discarded after delivery. Moreover, due to the fact that they are at the maternal—fetal interface, they become convenient for transplantation due to their low immunogenicity and immunomodulatory properties, making them a good candidate for allo- and xenotransplantation [27].

Despite the importance of *Canis familiaris* both as a patient and disease model, to our knowledge, in the literature, there are only general reviews of MSCs derived from foetal fluid and adnexa, based on research carried out on domestic animals [28,29,30]. The aim of the present review is to offer an update on the state of the art of canine MSCs isolated from foetal adnexa and fluid focusing on the findings in their clinical setting, when reported.

## 2. Stem Cells

The term stem cell was first coined in the nineteenth century by Edmund Beecher Wilson who used this term as a synonym for mitotically quiescent primordial germ cell. In 1963, Becker et al. [31] discovered that stem cells are undifferentiated cells able to perform self-renewal, as confirmed later by Weissman [32]. Based on their differentiation ability, stem cells are classified in totipotent, pluripotent and multipotent cells. Totipotent cells are able to differentiate in all cell lines, including extra-embryonic tissues; totipotent cells include zygote and the descendants of the first three cell divisions [33]. Pluripotent stem cells are embryonic stem cells derived from blastocysts ICM. These cells are able to propagate readily [34] and are capable of forming embryoid bodies that generate a variety of specialized cells, including neural, cardiac and pancreatic cells [35]. Despite their therapeutic potential, clinical use of ESCs presents, especially in human, ethical and application problems. Different authors have observed that after ESCs’ in vivo implantation, teratomas have been developed [36]. Another type of pluripotent stem cells is bioengineered IPS (Nelson et al. 2010) [37]. Utilizing IPS-based technology, all lineages of the adult body may become viable targets for replacement, avoiding immune intolerance. However, the unlimited differentiation potential of IPS is similar to ESCs, and thus the risk of dysregulated growth and teratoma formation requires stringent safeguards [37].

Many adult tissues, on the other hand, have multipotent stem cells within them, i.e., cells capable of producing a limited number of cell lines, appropriate to their location; these cells are named multipotent stem cells and the most studied among them are MSCs.

The most used mesenchymal tissues, sources of MSCs, are bone marrow and adipose tissue.

Mesenchymal stem cells are a population of multipotent stem cells which must meet the criteria established by ISCT: (i) plastic adherence when maintained in standard culture conditions; (ii) ability to differentiate in osteoblasts, adipocytes and chondroblasts when appropriately stimulated in vitro; (iii) expression of CD73, CD90 and CD105. On the contrary, MSCs lack expression of haematopoietic markers, such as CD14, CD34, CD45 and HLA-DR [38]. Due to these properties, MSCs offer a great chance for cell-based therapies and tissue-engineering applications.

## 3. Canine Mesenchymal Stem Cells from Foetal Fluids

### 3.1. Amniotic Fluid Mesenchymal Stem Cells

Amniotic fluid is characterized by a heterogeneous population of cells: cuboidal epitheloid cells, derived from foetal skin and urine, round cells, from foetal membranes and trophoblasts, and spindle-shaped fibroblastic cells, generated from mesenchymal tissues and are supposed to be the MSCs population of the AF [39]. Unlike in humans, in dogsm it is impossible to obtain AF using the amniocentesis technique. Moreover, due to its small volume, canine fluids provide a very small quantity of cells, which makes further cell expansion difficult. The use of the total recoverable volume of the AF of all foetuses improves the process of cells isolation, though there is no univocal agreement on which gestational period is the best in terms of cell yield. In 2011, Filioli-Uranio et al. [40] obtained fibroblast-like cells, with a DT of 1.12 ± 0.04 days, from AF recovered after hysterectomy in bitches between 25 and 40 days of pregnancy. These data are in contrast from those obtained by Fernandes et al. just one year later [41]. At earlier stage of gestation (25–40 days), Authors obtained fibroblast-like cells growing in adhesion but which failed to proliferate. Successful cells isolation was instead achieved from AF recovered at 50 days of gestation, as demonstrated by Choi et al. in 2013 [42]. Even if cell concentration decreased during the full-term stage of gestation [42], data recovered by Fernandes et al. [41] and Choi et al. [42] have shown the possibility of isolating AFMSCs after caesarean section in breeding bitches. As showed in Table 1, the expression of OCT4, CD44, DLA-DRA1, and DLA-79 by AFMSCs at passage P1 was observed by Filioli Uranio et al. [40], whereas from the next passage, cells expressed only CD44. On the contrary, Choi et al. [42] observed that at P5 of in vitro culture, AFMSCs express pluripotent stem cell markers OCT4, NANOG, and SOX2, as well as CD29 (β1 integrin), CD44, and CD90 (Thy1) [42]. These different results between different research groups could be determined both by the different gestational periods of samples’ recoveries and by different and not fully comparable techniques employed in stem cells characterization.

The hope of cell therapy as a new clinical approach to repair tissue damage relies on the characteristics of the mesenchymal stem cells, such as their low expression of polymorphic antigens that seems to enhance transplantation tolerance, making these cells useful for allotransplant and xenotransplant [70]. Filioli-Uranio et al. [40] observed a reduced expression of MHC genes in canine AFMSCs; indeed, only DLA-DRA1 and DLA-79 were expressed at P1, as previously demonstrated in human and other animal species [20,71,72,73].

Despite the differences found between the different research groups in the molecular characterization of AFMSCs, there is consensus on their differentiation potential. Indeed, it was demonstrated that these cells are able to differentiate in vitro into osteogenic, chondrogenic and adipogenic lineages [40,41,42], but also into neuronal [40,41,43] and hepatocyte-like cells [42]. Regarding neuronal differentiation, Fernandes et al. [41] observed that undifferentiated canine AFMSCs stained positively for nestin. Nestin is expressed in neural progenitor stem cells and pluripotent stem cells, which undergo neuronal differentiation, as well as in MSCs [74,75], including human AFMSCs [76]. Filioli-Uranio et al. [40] validated these data, observing that canine AFMSCs, cultured in neuronal induced medium, stained positively for nestin and showed the presence of Nissl bodies and a neuronal-like morphology, as confirmed later by Kim et al. [43]. The expression of neural-specific genes, such as NEFL, NSE, TUBB3, and the astrocyte-specific gene, GFAP, significantly increased in AFMSCs after neural induction [43].

Liver transplantation is a last resort for patients with end-stage liver disease. In a study [42], canine AFMSCs were induced to differentiate in hepatocyte like-cells by HGF (an endocrine or paracrine factor, essential for liver development), OSM (essential for the maturation of hepatocytes), nicotinamide, and dexamethasone (essential for the development of hepatogenic morphology through the suppression of cell division). AFMSCs that underwent hepatic differentiation did not show typical hepatocyte morphological changes, but they expressed genes critical for hepatocyte differentiation [42]. Indeed, after being induced to hepatocytic differentiation, canine AFMSCs were strongly positive for TAT, α1-AT, GS, and ALB, all markers of mature hepatocytes, as well as for hepatocyte-specific markers, including ALB and TAT [42]. Taken together, the results reported by different Authors suggest that canine AFMSCs have the capacity for multilineage differentiation and have the potential to be a source for cell-based therapies in canine models of hepatic disease, as well as having the potential capacity for clinical treatment of neuronal precursor-cell transplantation.

Although the studies referenced above demonstrate the potential for canine AFMSCs for clinical uses, to our knowledge, no paper on regenerative therapies based on canine AFMSCs exists in the literature.

### 3.2. Umbilical Cord Blood Mesenchymal Stem Cells

The umbilical cord is the channel that connects the fetus and the placenta, considered as a physiological and inherent part of the fetus during prenatal development [77]. In 1989 [78], umbilical cord blood, flowing in the vein, was found to be a rich and readily available alternative source of primitive and unspecialized mesenchymal stem cells, probably derived from the fetal liver or bone marrow [79]. Canine UCBMSCs were isolated for the first time by Lim et al. in 2007 [80]. However, characterization and in vitro differentiation of these cells were carried out only in 2009 [44], and the results obtained were confirmed later by different Authors [45,46,47]. In addition to the expression of stemness markers, shown in Table 1, and in vitro differentiation in the three lineages requested by ISCT [38], canine UCBMSCs showed basically neuronal associated protein markers under the undifferentiated condition. Indeed, undifferentiated canine UCBMSCs slightly expressed GFAP, Tuj-1, and NF160 neuronal cell protein markers but they did not express Nestin and MAP2 [44]. After growing in neuronal induction medium, canine UCBMSCs exhibited morphological changes [44,46], appearing as sharp, elongated bi- or tripolar cells with primary, secondary and multi-branched processes. When inducted with neuronal differentiation media, canine UCBMSCs showed positive expression patterns for Nestin, GFAP, Tuj-1, MAP2, NF160 and NeuN [44,46], showing a percentage of cells stained with antibodies specific for NeuN higher than that of adipose tissue [46]. Unlike cells deriving from canine amniotic fluid, for which there are no references on their clinical use, as showed in Table 2, canine UCBMSCs have been tested for regenerative therapy since 2007, when Lim et al. [80] applied these cells in induced spinal cord injured dogs. Authors observed evidence of functional and sensory improvement after allogenic UCBMSCs transplantation, even though no evidence of regeneration of spinal cord tissue by magnetic resonance imaging and histology was observed [80]. However, in this first study, new neuronal formation in the injured structures of the spinal cord was observed after UCBMSCs transplantation, as well as no additional damage to the experimentally injured spinal cord such as inflammatory responses [80]. Moreover, study [80] showed that transplantation of UCBMSCs resulted in recovered nerve function in dogs after a spinal cord injury, as confirmed recently by Park et al. [81] and Ryu et al. [46]. In the research of Park et al. [81] cells transplantation was carried out 12 h, 1 week and 2 week after spinal cord injury induction. Canine UCBMSCs transplanted one week after SCI significantly improved clinical signs, evaluated using the Olby and Tarlov scales. In all groups, the scores gradually increased after 2 weeks, and decreased after 3 weeks [81]. Better results were obtained later by Ryu et al. [46] transplanting cells with Matrigel, seven days after SCI, into the parenchyma of the spinal cord, near the lesion site or directly into the injury epicenter. Matrigel maintains the microenvironment and exerts effects such as rescuing dying cells, increasing cell proliferation, blocking inflammatory and cytotoxic cytokines, promoting neuronal differentiation [82,83]. As previously observed by Park et al. [81], transplanted cell survival is increased during the subacute phase of SCI, when the lesion is not fully developed and MSCs may act as neuroprotective agent. On the contrary, when cells were transplanted 2 weeks after SCI, when fibrosis has progressed, no significant improvement in patient clinical signs was observed [81]. Moreover, the lesion epicenter may not be a favorable site for cell survival due to the presence of phagocytes, which could have been the cause of the score decreasing 3 weeks after transplantation in the previous work [81]. Regarding the influence of canine UCBMSCs on inflammation, in all studies, COX-2 protein expression was significantly decreased [46,80,81], also compared with MSCs derived from other sources [46], playing an important role in proliferation, migration and differentiation of endogenous spinal cord-derived neural progenitor cells in SCI.

Canine UCMSCs also showed higher osteogenic potential compared with BMMSCs and WJMSCs, as shown by greater levels of ALP activity, an early osteoblastic marker [45]. All MSCs induced substantial in vivo bone formation and significant differences in the levels of bone formation promoted in vitro by the various MSCs were not observed [45], indicating that the osteogenic potential in vitro and in vivo can be slightly different for each type of MSCs. This can be explained by several factors, such as in vivo vascularization promoted by VEGF, secreted by MSCs in different quantities. In the study of Kang et al. [45] VEGF in vitro production was determined to investigate the ability of cells to promote vascularization. Authors observed that canine BMMSCs produced higher quantities of VEGEF compared with canine UCBMSCs; however, BMMSCs had a weaker osteogenic capability in vitro but no differences were determined in bone formation after MSCs transplantation in vivo and hematopoietic tissues were observed in histological section [45]. Moreover, MSCs may affect bone formation stimulating induction and migration of endogenous cells. Previous studies by the same research group revealed that cytokines released by canine UCBMSCs 1 day after implantation can enhance bone regeneration [87,88].

The reported mortality of AKI ranges from 47% to 61% in dogs [89]. Traditional AKI treatment includes fluid administration, monitoring urine output, use of diuretics, anti-nausea agent, gastroprotectors, phosphorus absorbent, antioxidants, sodium bicarbonate for metabolic acidosis, antidote for nephrotoxin, and antibiotics for infection [90]. After nephron disruption, it is difficult for patients to overcome the disease without the aid of dialysis or renal transplantation; these therapeutic approaches in dogs have considerable limitations, including difficulty in using them in smaller animals due to their low total blood volume, immunological problems and the low availability of donor kidneys [84,91]. Recently, Lee et al. injected twice canine UCBMSCs directly into the renal corticomedullary junction of dog with induced AKI, followed by intravenous administration of gentamycin and cisplatin [47]. For evaluating renal function blood BUN and creatinine were determined. BUN levels increased, owing to elevated urea reabsorption caused by prolonged renal retention due to a decreased glomerular flow rate [92]. Creatinine, a muscular metabolic product, is a more precise indicator of renal function than BUN. Serum creatinine higher than 10 mg/dl was associated with failure to recover from AKI [93]. In the study of Lee et al. serum BUN and creatinine levels decreased in dogs treated with UCBMSCs and renal excretory function improved [47]. In transplanted dogs, these serological findings were associated with moderate renal lesions, including necrosis of tubules and glomerulus and shedding of tubular epithelium cells; on the contrary, dogs treated with PBS exhibited global cystic change of tubular tissue and massive interstitial leukocyte infiltration [47]. Up to the end of the experiment, no mortality was recorded in dogs with induced AKI and subsequently treated with UCBMSCs, suggesting that for this pathology, MSCs could be an alternative and valid treatment.

## 4. Canine Mesenchymal Stem Cells from Foetal Adnexa

### 4.1. Placenta and Foetal Membranes

The potential for the clinical application of fetal stem cells from the human amniotic membrane was first described in 2011 by Parolini and Caruso [94]. Since, interest in this tissue as a source of MSCs has also developed in dogs. As shown in Table 1, canine AMMSCs expressed embryonic and MSCs markers, such as OCT4, CD44, CD184, and CD29 and could differentiate into neurocytes, osteocytes, adipocytes and chondrocytes based on cell morphology, specific stains, and molecular analysis [40,48,49,50,51,52]. Regarding the immunomodulatory properties of AMMSCs, Borghesi et al. [51] found low MHC-I expression and no MHC-II expression, giving the MSCs the potential to escape recognition by CD4+T cells [55]. Moreover, at any gestational time no interleukin IL-1, IL-2, IL-6 and IL-10 labeling in AMMSCs was observed [51].

To use stem cells safely, it is necessary to know if there are risks of genetic instability, which may lead to tumorigenesis. Recently, Cardoso et al. [50] and Borghesi et al. [51] did not observe tumor formation after injecting canine AFMSCs, demonstrating that these cells are safe for in vivo application.

As previously reported for human placenta derived MSCs [95], Long et al. [53] and Amorim et al. [54], using a cytokine array demonstrated that unstimulated and stimulated DPCs secrete a number of paracrine factors. VEGF and MCP-1 are implicated in both neuroprotection and angiogenesis [56,57], IL-6 is an immunomodulatory cytokine with in vitro and in vivo demonstrated neuroprotective capabilities [58], and IL-8 is an immunomodulatory cytokine that promotes angiogenesis [59].

Long et al. [53], in 2018, after 1 week of co-culture with a neuroblastoma cell line (SH-SY5Y cells), demonstrated that DPCs could induce the formation of complex neural networks in SH-SY5Y cells, increasing the number of branching points and total segments of neurites in culture. In 2020, Amorim et al. [54] speculated that based on their potent pro-angiogenic, neuroprotective, immunomodulatory properties, MSCs could be a promising therapy for canine inflammatory brain disease. IBD syndrome comprehends idiopathic disorders subdivided based on histopathology findings, i.e., granulomatous meningoencephalomyelitis, necrotizing meningoencephalitis, and necrotizing leukoencephalitis, encompass also in the term of meningoencephalomyelitis of unknown origin, presumed to be an autoimmune disease with a genetic predisposition [60], similarly to multiple sclerosis in human [61]. In vitro experiments revealed that rat neural cells exposed to OGD conditions and co-cultured with DPCs exhibited dose-dependent improvement in cell survival and ATP production compared to the vehicle, indicating the importance of these cells and cell dose in achieving neuroprotection [62]. In vivo observation in a rat stroke model reveals that animals treated with DPCs exhibited significantly fewer behavioral deficits, with improvements in motor and neurological impairments, as confirmed by histological data. These observations highlight the possible benefits of investigation into the efficacy of autologous transplant of DPCs in dog stroke or IBD patients, considering the lack of available treatment for ischemic and IBD injury in dogs.

### 4.2. Wharton’s Jelly or Umbilical Cord Matrix

Wharton’s Jelly is a mesenchymal connective tissue developed from extraembryonic mesoderm and placed between umbilical vessels. WJ binds and encases the umbilical vessels, protecting them from twisting and compression during gestation. Since 1990, WJ has been considered an important source of MSCs in humans [63,64]. However, in canines, WJ is difficult to separate from umbilical cord matrix due to the small size of umbilical cord; in this review we will refer to cells isolated from these sources as WJMSCs. MSCs were successfully isolated from canine WJ for the first time by Seo and collaborators in 2012 [65]. These cells showed a typical mesenchymal immunophenotype and the ability to differentiate in vitro [65], as confirmed lately by other research groups [45,48,66,67] (Table 1). Recently, Souza et al. [66] demonstrated that the optimum conditions for canine WJMSCs osteogenic differentiation is co-culture with PRP and DBM, associated with optimum ALP levels and high levels of osteocalcin gene, a late marker of osteogenesis, and osteopontin gene, an important factor in bone remodeling [68,69].

Recently, the metabolic profile of canine WJMSCs was investigated and compared with that of canine ATMSCs cultured under the basal conditions [67]. In order to detect the total ATP production rates in living MSCs, serial additions of oligomycin, a specific inhibitor of the mitochondrial ATP synthase, and of rotenone plus antimycin A, inhibitors of mitochondrial complex I and III, respectively, were performed automatically and stepwise. This metabolic assay allowed the Authors to evaluate the amount of ATP produced by OXPHOS and glycolysis, which represent the two main metabolic pathways responsible for ATP production in mammalian cells. ATMSCs and WJMSCs showed a different total ATP production rate, since OXPHOS and glycolytic pathways in foetal adnexa MSCs provided a higher amount of cellular ATP than in ATMSCs. The energy map of both MSC types corroborates an aerobic energy metabolism with more active OXPHOS and glycolytic pathways in WJMSCs than ATMSCs, although the latter two showed a higher mitoATP/glycoATP ratio than WJMSCs, which highlights a prevailing oxidative phenotype. In the same paper, the key parameters of mitochondrial function, directly measured by the cell respiration profile of MSCs, were also investigated. Both basal respiration and ATP turnover showed higher values in WJMSCs than in ATMSCs. Knowledge of the mitochondrial status and especially of some bioenergetics parameters such as the spare respiratory capacity, which guarantees a metabolic flexibility, may help to select the best candidates for transplantation studies.

Different researchers have demonstrated that canine WJMSCs have a higher neurogenic potential in vitro [46,48]. Ryu et al. [46] transplanted WJMSCs with matrigel into the spinal cord parenchyma near the induced lesion site or directly into the injury epicenter 7 days after injury. The Authors focused their attention on the survival and integration of allogenic MSCs in the injured spinal cord, demonstrating that allogenic WJMSCs could successfully survive in injured spinal cords where they integrated into host tissue without using immunosuppressive agents and also improved hind-limb function following SCI [46]. In particular, after WJMSCs transplantation, reduced levels of reactive astrogliosis and macrophage infiltration into the lesion epicenter were observed [46]. However, even though many WJMSCs can survive following transplantation, very few cells can differentiate into neural-like cells in vivo [46]. Indeed, most NF160- and NeuN-positive neurons in an injured spinal cord were derived from endogenous spinal cord-derived neural progenitor cells or preserved neurons, thanks to MSCs neuroprotection and anti-inflammatory effects. This is also confirmed by the rapid recovery of treated dogs after transplantation.

Due to their anti-inflammatory potential, WJMSCs have been used in intraarticular treatment of dogs subjected to surgical tibial plateau leveling osteotomy [85]. Taroni et al., in 2017, showed that a single postoperative intraarticular injection of allogeneic WJMSCs leads to a level of postoperative lameness and pain outcome after TPLO similar to those observed in animals treated with long-term NSAIDs systemic administration. WJMSCs anti-inflammatory potential could be due to factor secreted with Extracellular Vesicles, nanoscale cellular products containing RNA, protein, and lipids [96]. WJMSC EVs present a diameter of 125 nm, low buoyant density (1.1 g/mL), and expression of EV proteins Alix and TSG101. Functionally, EVs inhibited CD4pos T cell proliferation in a dose-dependent manner and TGF-b was present on EVs as latent complexes most likely tethered to EV membrane by betaglycan. These observations demonstrate that canine WJMSC EVs utilizes TGF-b and adenosine signaling to suppress proliferation of CD4pos T cell. WJMSCs EVs could also be responsible for the results registered by Yang et al. [86] after WJMSCs intravenous injection in dogs with congestive heart failure secondary to myxomatous mitral valve disease. Indeed, decreases in blood lymphocyte, monocyte, and eosinophil counts immediately after MSC injection were observed. However, further studies are needed on these topics for clinical WJMSCs application in different pathologic conditions.

## 5. Conclusions

The studies presented in this review offer authoritative views on markers expression and therapeutic potential of canine MSCs from foetal tissues and fluids. As reported in human and other animal species, also in dog these sources are easily available so MSCs may have an attraction compared to other established SCs in different clinical approaches. However, more in vitro study on their metabolism and clinical applications are needed to fully understand their properties and to establish the future clinical use in the treatment of various diseases.

## Figures and Tables

**Table 1 animals-11-02254-t001:** Canine foetal fluid and adnexa derived mesenchymal stem cells: potential differentiation in vitro and molecular characterization.

MSCs Source	Potential	References	Markers Positive Expression	References
AF	Osteogenic-Chondrogenic-Adipogenic	[40,41,42]	RT-PCR: OCT4; CD44; DLA-DRA1; DLA79	[40]
	Neurogenic	[40,41,43]	ICC: OCT4; NANOG; SOX2	[42]
	Hepatogenic	[42]	FACS: CD29; CD44; CD90	[42]
UCB	Osteogenic	[44,45,46,47]	FACS: CD44; CD73; CD90; CD105	[44,45,46,47]
	Chondrogenic-Adipogenic	[44,45,46,47]		
	Neurogenic	[44,46]	GFAP, Tuj-1,NF160	[44]
Placenta	Osteogenic-Chondrogenic-Adipogenic	[40,48,49,50,51,52]	RT-PCR: OCT4; CD44; CD184; CD29	[40,48,49,50,51,52]
	Neurogenic	[53,54]	FACS: OCT4; SOX2; CD73; CD90; CD105; MHC1	[51,52,55]
WJ	Osteogenic	[45,48,56,57,58,59,60,61,62,63,64,65,66,67,68,69]	FACS: CD44; CD73; CD90; CD105	[45,48,56]
	Chondrogenic-Adipogenic	[45,48,56,57,58,59,60,61,62,63,64,65,66,67,68,69]		
	Neurogenic	[46,48]	PCR: CD44; CD90	[67]

**Table 2 animals-11-02254-t002:** Canine foetal fluid and adnexa derived mesenchymal stem cells: clinical applications.

MSCs Source	Clinical Application	References
UCB	Bone Defect	[45]
	SCI	[46,80,81]
	AKI	[47]
Placenta	Ischemic Stroke	[84]
WJ	Bone Defect	[66]
	SCI	[46]
	TPLO	[85]
	CHF	[86]

## Data Availability

No new data were created or analyzed in this study. Data sharing is not applicable to this article.

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
