# Peer review of "Current Status on Canine Foetal Fluid and Adnexa Derived Mesenchymal Stem Cells"

_animals, 2021, doi:10.3390/ani11082254_

Round 1

Reviewer 1 Report

The authors of this review aimed to update the state of the art on MSCs derived from adnexa and fetal fluids, focusing, when present, on the few existing clinical data.

The review is interesting and overall well done but needs some improvements.

line 57: "familiaris" should be written with a lowercase initial.

In the paragraph 2.1 on line 73, DT should be expressed in full, doubling time, because this acronym was not mentioned earlier in the text.

Whenever possible, other acronyms should also be presented in full the first time they appear in the text with the acronym in parentheses. Check all the text and also check the acronyms of the cells which must be expressed uniquely eg. AD-MSC or ADMSC.

Line 82: Oct-4 should be written uniquely every time it appears in the text e.g. line 85.

Line 82: replace "at P1" with "at passage P1".

Line 83: replace "from the next step" with "from the next passage".

Lines 97-98: revise this sentence because the concept expressed was previously reported.

Page 3: report the acronyms of some factors presented in full; some then they are presented on page 4 eg. GFAP and NF160.

In paragraph 2.2 it is not necessary to put the acronyms WJ and UCB in brackets because they are already presented in the introduction.

line 161: replace "stained with NeuN" with "stained with antibodies specific for NeuN".

line 170: replace "this study" with "the study".

line 177: transplanting what with matrigel? specifies "transplanting cells".

lines 177, 179, 307: establish whether matrigel should be capitalized or not.

lines 241, 263, 266, 307, etc. put the period after "et al".

lines 468-470: complete citation 39 (Ryu et al.) in the references.

Author Response

Reviewer 1:

The authors of this review aimed to update the state of the art on MSCs derived from adnexa and fetal fluids, focusing, when present, on the few existing clinical data.

The review is interesting and overall well done but needs some improvements.

line 57: "familiaris" should be written with a lowercase initial.

The authors made the correction suggested by the reviewer

In the paragraph 2.1 on line 73, DT should be expressed in full, doubling time, because this acronym was not mentioned earlier in the text.

The list of abbreviation has been added at the beginning of the manuscript as suggested by the reviewer.

Whenever possible, other acronyms should also be presented in full the first time they appear in the text with the acronym in parentheses. Check all the text and also check the acronyms of the cells which must be expressed uniquely eg. AD-MSC or ADMSC.

Line 82: Oct-4 should be written uniquely every time it appears in the text e.g. line 85.

The authors made the correction suggested by the reviewer

Line 82: replace "at P1" with "at passage P1".

The authors made the correction suggested by the reviewer

Line 83: replace "from the next step" with "from the next passage".

The authors made the correction suggested by the reviewer

Lines 97-98: revise this sentence because the concept expressed was previously reported.

The authors made the correction suggested by the reviewer

Page 3: report the acronyms of some factors presented in full; some then they are presented on page 4 eg. GFAP and NF160.

In paragraph 2.2 it is not necessary to put the acronyms WJ and UCB in brackets because they are already presented in the introduction.

The list of abbreviation has been added at the beginning of the manuscript as suggested by the reviewer.

line 161: replace "stained with NeuN" with "stained with antibodies specific for NeuN".

The authors made the correction suggested by the reviewer

line 170: replace "this study" with "the study".

The authors made the correction suggested by the reviewer

line 177: transplanting what with matrigel? specifies "transplanting cells".

The authors made the correction suggested by the reviewer

lines 177, 179, 307: establish whether matrigel should be capitalized or not.

The authors made the correction suggested by the reviewer

lines 241, 263, 266, 307, etc. put the period after "et al".

The authors made the correction suggested by the reviewer

lines 468-470: complete citation 39 (Ryu et al.) in the references.

The authors realized that this citation was a repetition of the number 37, so it was replaced throughout the entire manuscript. After revision the new number of this reference is 57

Reviewer 2 Report

In this work, the authors explore the most recent evidence regarding the importance of fetal fluids and fetal adnexa in dogs for obtaining mesenchymal cells to be directed towards the most diverse cell types for therapeutic purposes in companion dogs.

The review is well structured, and the bibliography is sufficiently recent so as to be able to photograph the state of the art on the subject up to the present.

The topic is interesting as it highlights the possibility of obtaining experimental material (liquids and fetal adnexa) from the sterilizations of the bitches which are routinely carried out in veterinary clinics and hospitals. This allows to obtain rigorous experimental protocols, without the use of experimental animals and exploiting biological parts destined for destruction.

I would recommend a revision of the English in order to make the sentences a little shorter thus increasing the readability and comprehensibility of the text.

Author Response

REVIEWER 2

In this work, the authors explore the most recent evidence regarding the importance of fetal fluids and fetal adnexa in dogs for obtaining mesenchymal cells to be directed towards the most diverse cell types for therapeutic purposes in companion dogs.

The review is well structured, and the bibliography is sufficiently recent so as to be able to photograph the state of the art on the subject up to the present.

The topic is interesting as it highlights the possibility of obtaining experimental material (liquids and fetal adnexa) from the sterilizations of the bitches which are routinely carried out in veterinary clinics and hospitals. This allows to obtain rigorous experimental protocols, without the use of experimental animals and exploiting biological parts destined for destruction.

I would recommend a revision of the English in order to make the sentences a little shorter thus increasing the readability and comprehensibility of the text.

The Authors thank the reviewer for the comments.

As suggested by the reviewer, a revision of the English of the manuscript was made.

Reviewer 3 Report

The manuscript under evaluation is a review of Current Status on Canine Foetal Fluid and Adnexa Derived Mesenchymal Stem Cells. It is an important and current topic in veterinary and human medicine perspectives.

My comments have to do with the lesser or greater difficulty in reading depending on the type and training of readers, the theme and because it is only presented in plain text without figures and/or tables.

Comment 1: Due to the nature of the manuscript and to facilitate reading at any time, I think that a list of Abbreviations at the beginning of the manuscript is justified.

Comment 2: In the Introduction I think it would be interesting in a sentence to refer to the theme for other animal species/of veterinary interest. For example:

Iacono E, Rossi B, Merlo B. Stem cells from foetal adnexa and fluid in domestic animals: an update on their features and clinical application. Reprod Domest Anim. 2015; 50(3):353-64. doi: 10.1111/rda.12499.

Ambrósio CE, Orlandin JR, Oliveira VC, Motta LCB, Pinto PAF, Pereira VM, Padoveze LR, Karam RG, Pinheiro AO. Potential application of aminiotic stem cells in veterinary medicine. Anim Reprod. 2020; 16(1):24-30. doi: 10.21451/1984-3143-AR2018-00124.

Cremonesi F, Corradetti B, Lange Consiglio A. Fetal adnexa derived stem cells from domestic animal: progress and perspectives. Theriogenology. 2011; 75(8):1400-15. doi: 10.1016/j.theriogenology.2010.12.032. PMID: 21463720.

Comment 3: In the Introduction I think it would be interesting in a sentence to mention some of the pathologies that can benefit from this approach.

Comment 4: In the Introduction I think it would be interesting to make a brief reference to whether isolation protocols differ between species and whether they can affect their success.

Comment 5: I think it would be interesting, after the Introduction, a brief chapter, to frame any reader, about the basic concepts of stem cells, definition and characteristics of the stem cells, criteria of classification, according to their potential for differentiation, etc…

Comment 6: Line 73, please explain DT

Comment 7: Line 222, please explain BUN

Comment 8: To make the manuscript lighter, less dense, what do the authors think about making a table of this type? The readers were soon with a photograph of the theme and its applications. And maybe also a Figure.

Tabela X: Canine foetal fluid and adnexa derived mesenchymal stem cells and their potential

Derived stem cells

Potential

Markers

Ref

Amniotic Fluid Mesenchymal

Or say they appear in the text

Umbilical Cord Blood Mesenchymal

osteogenic potential…

Placenta and Foetal Membranes

Wharton’s Jelly or Umbilical Cord Matrix

Author Response

REVIEWER 3

The manuscript under evaluation is a review of Current Status on Canine Foetal Fluid and Adnexa Derived Mesenchymal Stem Cells. It is an important and current topic in veterinary and human medicine perspectives.

My comments have to do with the lesser or greater difficulty in reading depending on the type and training of readers, the theme and because it is only presented in plain text without figures and/or tables.

Comment 1: Due to the nature of the manuscript and to facilitate reading at any time, I think that a list of Abbreviations at the beginning of the manuscript is justified.

The list of abbreviation has been added at the beginning of the manuscript as suggested by the reviewer.

Comment 2: In the Introduction I think it would be interesting in a sentence to refer to the theme for other animal species/of veterinary interest. For example:

Iacono E, Rossi B, Merlo B. Stem cells from foetal adnexa and fluid in domestic animals: an update on their features and clinical application. Reprod Domest Anim. 2015; 50(3):353-64. doi: 10.1111/rda.12499.

Ambrósio CE, Orlandin JR, Oliveira VC, Motta LCB, Pinto PAF, Pereira VM, Padoveze LR, Karam RG, Pinheiro AO. Potential application of aminiotic stem cells in veterinary medicine. Anim Reprod. 2020; 16(1):24-30. doi: 10.21451/1984-3143-AR2018-00124.

Cremonesi F, Corradetti B, Lange Consiglio A. Fetal adnexa derived stem cells from domestic animal: progress and perspectives. Theriogenology. 2011; 75(8):1400-15. doi: 10.1016/j.theriogenology.2010.12.032. PMID: 21463720.

See the paper at line 154-158

Comment 3: In the Introduction I think it would be interesting in a sentence to mention some of the pathologies that can benefit from this approach.

See the paper at line 118-1139.

Comment 4: In the Introduction I think it would be interesting to make a brief reference to whether isolation protocols differ between species and whether they can affect their success.

The methods used for MSCs isolation are similar in different species and frequently related to the procedures adopted in the laboratory where the research is carried out. Sometimes the media employed for cell culture are different, depending on cell growth and glucose uptake. However, the present review is focused on the characterization and clinical applications of canine MSCs derived from foetal fluids and adnexa while in literature are present other reviews focused on different aspect on cell culture in domestic animals. For these reasons the Authors consider it redundant and superfluous to deal with this in the present reviews.

Comment 5: I think it would be interesting, after the Introduction, a brief chapter, to frame any reader, about the basic concepts of stem cells, definition and characteristics of the stem cells, criteria of classification, according to their potential for differentiation, etc…

See the paper at line 160-185; a brief chapter on stem cells has been added, as suggested by the reviewer.

Comment 6: Line 73, please explain DT.

The list of abbreviation has been added at the beginning of the manuscript as suggested by the reviewer.

Comment 7: Line 222, please explain BUN.

The list of abbreviation has been added at the beginning of the manuscript as suggested by the reviewer.

Comment 8: To make the manuscript lighter, less dense, what do the authors think about making a table of this type? The readers were soon with a photograph of the theme and its applications. And maybe also a Figure.

Table 1 and 2 have been added to the manuscript ad suggested by the reviewer.